# A Brief Overview of the Microstructural Engineering of Inorganic–Organic Composite Membranes Derived from Organic Chelating Ligands

**DOI:** 10.3390/membranes13040390

**Published:** 2023-03-30

**Authors:** Sulaiman Oladipo Lawal, Masakoto Kanezashi

**Affiliations:** Chemical Engineering Program, Graduate School of Advanced Science and Engineering, Hiroshima University, 1-4-1 Kagamiyama, Higashi-Hiroshima 739-8527, Japan

**Keywords:** organic chelating ligands, microstructure engineering, composite membranes, crosslinking, network modification, network formation

## Abstract

This review presents a concise conceptual overview of membranes derived from organic chelating ligands as studied in several works. The authors’ approach is from the viewpoint of the classification of membranes by matrix composition. The first part presents composite matrix membranes as a key class of membranes and makes a case for the importance of organic chelating ligands in the formation of inorganic–organic composites. Organic chelating ligands, categorized into network-modifying and network-forming types, are explored in detail in the second part. Four key structural elements, of which organic chelating ligands (as organic modifiers) are one and which also include siloxane networks, transition-metal oxide networks and the polymerization/crosslinking of organic modifiers, form the building blocks of organic chelating ligand-derived inorganic–organic composites. Three and four parts explore microstructural engineering in membranes derived from network-modifying and network-forming ligands, respectively. The final part reviews robust carbon–ceramic composite membranes as important derivatives of inorganic–organic hybrid polymers for selective gas separation under hydrothermal conditions when the proper organic chelating ligand and crosslinking conditions are chosen. This review can serve as inspiration for taking advantage of the wide range of possibilities presented by organic chelating ligands.

## 1. Introduction

The separation, concentration and purification of material mixtures is a major challenge facing industries such as the chemical, petrochemical, petroleum, textile, food and pharmaceutical industries [1]. In recent decades, attention towards membranes and membrane separation processes has grown significantly [2] because membranes provide simple, cost-effective and energy-saving alternatives to state-of the-art techniques [3,4]. A membrane in the general sense refers to a “thin”, selectively permeable, or semi-permeable barrier or boundary that allows some material species such as molecules, ions or other similarly sized particles to pass through it and prevents the passage of others. Separation membranes serve as a third phase, creating a barrier between two bulk phases. The aim of a membrane process is to concentrate a component in a phase into another phase through the permeable barrier via mass transport [5]. Separation membranes may be categorized by their material make-up, physical configuration, morphology and material composition. An overview of this categorization is shown in Figure 1.

The material make-up of a membrane refers to the chemical nature of its constituting matter, which can provide a reasonable basis for categorization. Since matter can be either organic or inorganic, membranes can be generally classified by their material make-up into two large categories: polymeric and inorganic membranes. Several works have reported membranes from these two categories [6,7,8,9,10,11]. Inorganic membranes can be further categorized as follows: (i) microporous inorganic membranes and (ii) dense inorganic membranes. Examples of microporous inorganic membranes include amorphous silica [12,13,14,15] and zeolites (which contain silicon, aluminum and oxygen atoms arranged into a lattice structure) [16,17,18,19]. Dense inorganic membranes, on the other hand, include dense metal membranes that can achieve exceptionally high H_2_ selectivities, and they typically include glass or special metals and their alloys such as palladium [10], niobium, tantalum, vanadium, nickel, platinum [20], Pd-Au, Pd-Ag, Pd-Cu [10], Pd-Ta [21], Pd-Ru, Pd-Nb, Pd-Mo, and Pd-Cu-Ag [22].

Membranes can be further classified according to the manner in which the membrane module has been physically configured for application. Polymeric and inorganic membranes may be configured in various ways that are not particular to a membrane material, although a configuration may be more suited to one membrane material over another. Common configurations include flat-sheet membranes [8,23,24], hollow-fiber membranes [25,26,27,28] and tubular membranes. Membranes may also have different morphologies depending on their structure. Based on membrane morphologies, membranes may be further categorized as symmetric membranes (also referred to as isotropic membranes) [29] and asymmetric membranes (also referred to as anisotropic membranes) [30].

### 1.1. Separation Membrane Classifications by Matrix Composition

Although separation membranes have been commonly classified from the perspectives of their material make-up, morphology and physical configuration as discussed above, a new classification of membranes has become necessary due to the evolution of membrane separation technology to meet new targets [31,32,33]. A new generation of membranes has been developed to obtain better performance metrics (selectivity, permeability, cost, handling, etc.) compared to those of the previous generation of membranes. Such evolutionary methodologies involve the combination of different matrices to harnessing the strengths of each matrix for improved performance. Thus, a new holistic classification is suggested, as illustrated in Figure 2.

#### 1.1.1. Single-Matrix Membranes

This is the most basic form of a membrane matrix—a membrane comprising only one type of material, i.e., a single-component membrane (Figure 2a). Many of the popular membrane types fall under this category as they serve as the bases for further developments. Typically, single-matrix membranes will include all the various types of pure polymer membranes (polydimethoxysilane, polysulfone, polyimide, polyetherimide, cellulose acetate, etc.), pure metal oxides and silica membranes (silica, zirconia, titania, alumina, etc.), pure metals (palladium, nickel, tantalum, molybdenum, niobium, etc.) and zeolites (ZIF, SAPO, AlPO_4_, and MeAPO structures, among others).

#### 1.1.2. Mixed-Matrix Membranes

Mixed-matrix membranes (MMM) are a combination of polymer and inorganic matrices. A pioneering work on MMMs was published by Koros and coworkers [34]. The concept of mixed-matrix membranes was introduced to overcome the limitations of both polymer membranes (viz., trade-offs, physical aging and plasticization) and inorganic membranes (viz., brittleness and the cost of inorganic processing) [35,36,37]. Typically, an MMM consists of a bulk polymer phase into which the inorganic phase is dispersed [36]. The inorganic phase dispersed into the polymer bulk phase, as shown in Figure 2b, is referred to as a filler. The inorganic fillers traditionally incorporated can be categorized as zeolitic and non-zeolitic fillers [37]. The zeolitic fillers include conventional zeolites, and AlPO and SAPO molecular sieves, whereas non-zeolitic fillers include carbon molecular sieves (CMS), porous and dense silica nanoparticles, metal oxide nanoparticles [37], activated carbon, metal–organic frameworks (MOFs), carbon nanotubes (CNTs), mesoporous materials and lamellar inorganic materials [35,36].

#### 1.1.3. Doped Matrix Membranes

A membrane can acquire new characteristics and properties that improve its performance by introducing into it a metal or non-metal ion. Such an addition is referred to as doping. This strategy has been especially used in silica and silica-based membranes [38,39,40,41]. Doping a material with an anion or cation is used to direct a membrane’s microstructure or functionality towards developing an enhanced performance. For example, fluorine as an anion has been used to add a functionality to an organosilica membrane for improved propylene/iso-butane separation [41], for the improved hydrothermal stability of a silica membrane [42,43] or for microstructural control of the membrane pore size [40]. Palladium doping of a silica membrane has been used to improve hydrogen permselectivity [44]. Nickel doping of aminosilica has also been recently studied [45,46]. Figure 2c shows the idealized structure of an ion-doped matrix, which was realistically confirmed in a Pd-doped silica membrane as studied by Kanezashi et al. and is shown in Figure 3 [47].

#### 1.1.4. Composite Matrix Membranes

In mixed-matrix membranes [37] and doped matrix membranes [47], fillers or dopants are present as separate entities in the bulk of the primary matrix. In mixed-matrix membranes, the challenge of inorganic particle agglomeration in the polymer matrix, especially in glassy polymers, is a major concern [36]. In most doped matrix membranes (except for Pd-doped membranes for H_2_ permselectivity), the dopants are not present as separate matrices but as functionality or microstructural change agents. For composite matrix membranes, however, two separate moieties are integrated into one structure and act as a single matrix for active separation. Furthermore, the different structures comprising a composite matrix can act as structural, functional or molecular-sieving agents, or a combination of these. These composites can be classified into two major categories: inorganic–inorganic and inorganic–organic composites.

Inorganic–inorganic composites

Inorganic–inorganic composites include several ceramic composites such as SiO_2_-ZrO_2_ [48,49,50], TiO_2_-ZrO_2_ [51,52,53] and SiO_2_-TiO_2_ [54]. SiO_2_-ZrO_2_ composite membranes have been used for pervaporation and nanofiltration applications [55,56,57]. The hydrogen separation performance of SiO_2_-ZrO_2_ hybrid membranes fabricated through chemical vapor deposition (CVD) has also been investigated [50]. Figure 4 shows the cross-sectional morphology of typical supported TiO_2_-ZrO_2_ membrane fabricated by Anisah et al. [51].

Inorganic–organic composites

Inorganic–organic composites are diphasic hybrids that can be grouped into two large categories: class I and class II inorganic–organic hybrids. A class I hybrid is such that the organic and inorganic matrices are linked by weak interactive forces, such as hydrogen and van der Waals bonding [58]. An example of class I hybrids is the interpenetrating polymer and inorganic networks (IPNs) expressed in the schematic image in Figure 5a [59]. In class II hybrids, on the other hand, the interaction between the inorganic and organic parts, which includes strong bonds such as covalent and ionocovalent bonds, serve to improve the inorganic matrix qualities including improved hydrothermal stability [60], pore size controllability, [61] targeted functionalization, [62], etc. Examples include organically modified transition-metal oxides/ceramics [63], metal–organic frameworks (MOFS) [64,65], organosilica materials (-Si-R-Si-), etc. [61,62]. Class II hybrids, as expressed in organosilica membranes, are shown in the schematic image of Figure 5b [61].

### 1.2. Transition-Metal Alkoxides and Organic Chelating Ligands (OCLs)

The primary advantage of sol–gel synthesis is the ability to obtain materials with predetermined characteristics based on the experimental conditions applied. In addition to this, the synthesis of inorganic–organic hybrid materials is easily achieved through the introduction of organic functionalities in the sol–gel synthesis procedure [66]. One way of adding an organic functionality is to chemically modify a precursor prior to sol–gel synthesis [67]. A good example of chemically modifying precursors is found in the use of coordination ligands to form coordinate complexes, where the precursor to be modified is a transition-metal ion compound. Pioneering studies on the chemical modification and sol–gel chemistry of transition-metal alkoxides were carried out by Sanchez, Livage and their coworkers [68,69]. They studied topics that included the partial charge model, gelation, various organic coordination ligands, alcohol interchange, etc. A few important aspects are elaborated on in the following sections.

#### 1.2.1. Transition-Metal Alkoxides

A transition-metal alkoxide has a transition metal bonded to an alkoxyl group (OR; R= alkyl group). The most common transition-metal alkoxides that are encountered in sol–gel chemistry, catalysis and membrane science are alkoxides of zirconium and titanium, which serve as precursors for ZrO_2_ and TiO_2_, respectively. These metal alkoxides are much more reactive than the equivalent silicon alkoxide (Si(OR)_4_). For example, the hydrolysis rate of Ti(OEt)_4_ is about five times more than that of Si(OEt)_4_. This is because the electronegativities of the transition-metal atoms are much smaller than those of silicon, and thus, their positive partial charge is higher [68].

Another major characteristic of transition-metal alkoxides is coordination expansion [69]. This is the tendency of positively charged metal atoms to increase their coordination number by using their vacant orbitals to accept electrons from nucleophilic ligands. Their reaction with molecules of the form XOH containing reactive hydroxyl groups follow the general format in Equation (1) [69].
M (OR)*_n_* + *x*HOX → M (OR)*_n−x_* (OX)*_x_* + *x*ROH(1)

Such a reaction can correspond to hydrolysis when X is a hydrogen atom H, a condensation reaction when X is the same or another metal M, or a chemical modification when X is an organic group R’.

#### 1.2.2. The Partial Charge Model

According to the partial charge model, the electronegativity *χ*_i_ of an atom varies linearly with its partial charge, and the electron transfers between atoms that come together to form a molecule must stop when all electronegativities approach a mean electronegativity *χ* [70]. Therefore, it follows that the metal atom and the leaving group ROH have to be positively charged. The nucleophilic substitution in transition-metal alkoxides can be explained by the knowledge of the charge distribution over the atoms [68].

#### 1.2.3. Gelation Time

Stockmayer [71] proposed an equation to estimate the gelation time when metal alkoxides undergo nucleophilic substitution. The gelation time *t_g_* depends on the initial concentration of the alkoxide, *c*_0_, the functionality, *f* (the number of OR groups that can be substituted upon hydrolysis), and the bimolecular rate constant of polymerization, *k,* as shown in Equation (2).
(2)$$t_{g}={\left[c_{0}k\left(f^{2}-2f\right)\right]}^{-1}$$

When only alkoxides are used (to form SiO_2_-ZrO_2_ or TiO_2_-ZrO_2_ composites, for example), *k* becomes the condensation rate constant. Furthermore, functionality can be well defined if hydrolysis proceeds without condensation. Therefore, Equation (2) applies only when hydrolysis occurs well in excess of condensation reactions. In any other case, when the rate of condensation is equal to or higher than the rate of hydrolysis, precipitates or colloidal gels form, and aggregates cannot be neglected [68].

#### 1.2.4. Organic Chelating Ligands

Most transition-metal alkoxides are very reactive in hydrolysis and condensation reactions, leading to particle agglomeration and segregation when forming composite oxides. As such, the need to stabilize or slowdown their reaction rates when forming composite oxides arises [66,68]. This is achieved by chemically modifying the alkoxides with complexing agents that are usually hydroxylated nucleophilic ligands such as carboxylic acids and β-diketones. For example, acetylacetone AcAc (CH_3_-CO-CH_2_-CO-CH_3_), an organic ligand in its enolic form, contains hydroxyl groups which can react with metal alkoxides. According to the partial charge theory, the acetylacetonate group ACAC^-^ (CH_3_-CO-CH_2_-CO-CH_2_) reacts with a titanium tetrapropoxide to give the oligomeric species Ti (OPr)_3_ (CH_3_-CO-CH_2_-CO-CH_2_)—structurally depicted in Figure 6—and favors the release of the alcohol PrOH (propanol) [68]. Such nucleophilic reactions between hydroxylated ligands and transition-metal alkoxides have been studied extensively [72,73,74] and are referred to as chelation.

Since Hoebbel and coworkers [75] found that some of the ligands chelated to transition metals are hydrolytically stable to a very good extent in the range of a period of a week under continuous stirring in the presence of water, the adoption of organic chelating ligands to form inorganic–organic composite membranes has become possible.

## 2. Organic Chelating Ligand-Derived Inorganic–Organic Composites

As previously discussed, the interaction between the inorganic and organic parts in class II inorganic–organic hybrids involves strong bonds such as covalent and ionocovalent bonds. Recognizing the unique chemistry between transition-metal ions and organic chelating ligands, researchers have been able to develop a new class of inorganic–organic composite membranes that have shown promise in gas separation and filtration applications. As also pointed out earlier, the original intention of OCLs is to control the hydrolysis reaction rates of transition-metal alkoxides when forming composites, especially with silica. However, if the chelation bond between the OCL and transition-metal alkoxide can be preserved, OCL-derived inorganic–organic composites can thus be classified as class II hybrids. Therefore, the unique difference between OCL-derived inorganic–organic composites and other members of the class II hybrids is that they are formed from individual building blocks, as shall be elaborated on later. The versatility of the chemical structures and special functions of OCLs makes it essential to explore their network engineering effects.

### 2.1. Organic Chelating Ligands as Molecular Engineering Agents

Complexation of alkoxides with nucleophilic chelating ligands allows for the design of molecular precursors and the sol–gel synthesis of tailor-made materials [68]. Consequently, the role of a chelating ligand in engineering molecular-sieving membrane networks can be predetermined according to two major functions: as a network modifier or network former [76] (Figure 7).

#### 2.1.1. Network Modifiers

When the chelating ligand is a simple non-hydrolysable group, it will have a network-modifying effect [77] (Figure 7). In this case, the ligand can create a physically or chemically different site [76]. Organic chelating ligands that fall into this category include acetylacetone, acetic acid, isoeugenol, ethyl acetoacetate, etc., and are depicted in Figure 8a.

#### 2.1.2. Network Formers

An organic chelating ligand may bear a reactive group that can, for example, polymerize or copolymerize to form porous polymer composites [76,77], as illustrated in Figure 7. The ligand can be said to be a network former in this case. Organic chelating ligands that can serve as network formers include allyl acetoacetate, methacryloxyethyl acetoacetate, methacrylic acid, etc., as shown in Figure 8b. The common feature of these ligands, in addition to the hydroxylated nucleophilic species they bear, is the unsaturated C=C bonds that can be utilized as crosslinking agents.

Despite the first research into the chemistry of transition metals and chelating ligands occurring in the 1940s [71,78] and 1960s [70], applications with composite ceramic membranes are relatively new, and one of the earliest appeared in 2001 and was carried out by Benfer and coworkers [79], who studied the preparation of TiO_2_ and ZrO_2_ nanofiltration membranes modified with acetylacetone, acetic acid and diethanolamine. Considering the numerous amounts of these ligands and their unique chemistries, there seems to be wide scope of possibilities when utilizing organic chelating ligands and composite ceramic membranes.

### 2.2. General Structural Building Blocks of Organic Chelating Ligand-Derived Inorganic–Organic Composites

Figure 9 shows a general framework for the fabrication of OCL-derived inorganic–organic composites that is based on the work of Haas [80]. This framework is composed of four structural elements or building blocks: a siloxane network, a transition-metal oxide network (TMO) that provides chelation sites, an organic modifier and crosslinking of the organic modifiers. R or (R) represents an organic modifier that can be a ligand for the modification of the TMO or organic group bonded to the siloxane. As discussed earlier, these organic groups may serve as network modifiers or network formers. As for the siloxane element, R will have a network-modifying effect when organic groups attached to the silane precursor are non-hydrolysable, such as a hydrophilic group (aminopropyltrimethoxysilane), a hydrophobic group (phenyltriethoxysilane) or a combination of hydrophobicity and oleophobicity (tridekafluoro-1,1,2,2-tetrahydrooctyltriethoxysilane). The silane precursor may also possess a reactive group, such as a polymerizable unit, which will serve as a network former in this case [81]. For network formation, precursors such as methacryloxypropyltrimethoxysilane and vinyltrimethoxysilane can be utilized in conjunction with network-forming organic chelating ligands bonded to the transition-metal oxide [80,81]. Therefore, organic groups serving as network modifiers or network formers may be bonded to either the siloxane or TMO backbone.

### 2.3. General Preparation Routes of Organic Chelating Ligand-Derived Inorganic–Organic Composites

OCL-derived inorganic–organic composites may be formed based on all or part of the structural building blocks presented in Figure 9 depending on the applications. In general, the transition-metal alkoxide is modified by the OCL first, which is followed by the hydrolysis and polycondensation of the silane and the OCL-modified transition-metal alkoxide in the presence of a catalyst to form the inorganic composite network of Si and the transition metal in -Si-O-M-OCL linkages. OCLs differ in their hydrolytic stability, and thus, the amount of the ligand chelated to the transition metal that remains after the hydrolysis reaction will differ from ligand to ligand [75]. The amount of water utilized in the hydrolysis reaction also affects the concentration of ligands in the resulting composite. Therefore, an optimal choice with regard to ligand type and water amount must be chosen to give the best results [75]. Hydrolysis and polycondensation are then followed by crosslinking of the reactive organic moieties in the presence of heat or light to form the organic network, as depicted in Figure 10 [82]. In the case where crosslinking of the organic modifiers does not apply due to the ligands having no reactive units (such as those shown in Figure 8a), the organic moieties only serve to modify the network, leading a composite made of only three of the four structural elements.

## 3. Inorganic–Organic Composites and Membranes from Network-Modifying Ligands

Figure 11 shows the typical preparation flow of inorganic–organic composites from pure network-modifying ligands. As pointed out earlier, the first step usually involves the chelation of the transition-metal alkoxide, which forms part of the ceramic network. This is then followed by co-hydrolysis and polycondensation reactions in the presence of water and acid catalysts to form the hybrid composite network. Ligand-modified composite membranes have shown a better molecular-sieving performance compared to the corresponding unmodified composites with the same mixed oxide composition. This observation has led to more applications of ligand-modified composite ceramic membranes in gas separation and filtration. Table 1 summarizes these membranes and their applications. The theory of the improved molecular sieving of ligand-modified composite membranes is that the chelating ligands coordinated to the transition-metal centers modify the intraparticle network pores by physically occupying the networks and serving as a shield against the transport of large molecules, as illustrated in Figure 11. Fukumoto et al. provided an insight into the molecular-sieving roles of organic chelating ligands by changing the ligand/alkoxide molar ratio and found that by increasing the ligand content, the molecular-sieving performance was improved [83].

### 3.1. Applications for Gas Separation

In certain studies, researchers have replaced silica as a core structural element of OCL-derived inorganic–organic composites with pure transition-metal oxide composites, especially a composite of TiO_2_ and ZrO_2_. This is conducted to increase the concentration of OCLs in the networks where the chelating ligand is bonded to both the transition metals within the composite. Fukumoto et al. prepared isoeugenol- and diethanolamine-modified TiO_2_-ZrO_2_ composite membranes that showed superior molecular-sieving selectivities compared to unmodified TiO_2_-ZrO_2_ composite membranes [83]. Elsewhere, Spijksma and coworkers also showed a diethanolamine-modified TiO_2_-ZrO_2_ composite membrane with a H_2_ permeance and H_2_/butane selectivity of 3.0 × 10^−7^ mol m^−2^ s^−1^ Pa^−1^ and 54, respectively [84].

The molecular-sieving properties of an acetyl acetone (AcAc)-modified SiO_2_-ZrO_2_ membrane were also studied by our group [87]. In a similar fashion to Fukumoto et al. [83], it was observed that the AcAc-modified SiO_2_-ZrO_2_-derived membrane showed a molecular-sieving performance with a H_2_ permeance of 9.9 × 10 ^−7^ and H_2_/SF_6_ permeance ratio of 7600, which was better than the results when using a pure SiO_2_-ZrO_2_-derived membrane (H_2_ permeance: 1.4 × 10 ^−6^ mol m^−2^ s^−1^ Pa^−1^; H_2_/SF_6_ permeance ratio: 11), as shown in Figure 12.

### 3.2. Applications for Nanofiltration

Organic chelating ligands used in the modification of ceramic composites, depending on their chemistry, also serve the purpose of altering the functionality of such ceramic networks such as hydrophilicity/hydrophobicity, oleophobic properties, CO_2_ solubility, etc. Sada et al. modified TiO_2_-ZrO_2_ with ethyl acetoacetate and 2,3-dihydroxynaphthalene organic chelating ligands to achieve nanofiltration membranes with high water permeabilities of 5.6 and 9.2 (L m^−2^ h^−1^)/bar, respectively, while maintaining a high rejection of acid red [88].

Iesako and coworkers studied the effects of 3,5-di-tert-butylcatechol (DTBC) on the pore size and surface property control of TiO_2_-ZrO_2_ composite membranes for nanofiltration [89]. In this study, it was found that the addition of DTBC not only generated a membrane with a narrower pore size, but the membrane also showed a hydrophobic nature. Fabricated at 300 °C in a nitrogen atmosphere, the TiO_2_-ZrO_2_-DTBC membrane displayed a narrow pore size and hydrophobic surface, allowing for a low-molecular-weight cut-off (MWCO) of 500 g/mol. After calcination in air at 350 and 500 °C to remove the intra-network DBTC, both the pore size and hydrophilicity of the membranes increased. Thus, residual DTBC in the TiO_2_-ZrO_2_ network was responsible for the narrow pore size and high hydrophobicity of the membranes. This goes to show that the modification via organic chelating ligands works the same in nanofiltration and gas separation.

## 4. Inorganic–Organic Composites and Membranes from Network-Forming Ligands

As discussed in Section 2.3, certain preparation procedures of OCL-derived inorganic–organic composites do not require the crosslinking of the OCLs and other organic modifiers when they do not possess reactive units. However, after hydrolysis and condensation of OCL-modified precursors bearing reactive units such as polymerizable carbon bonds, crosslinking or curing follows in the presence of heat or UV light and a radical initiator/curing agent such as azobisisobutyronitrile and dibenzoyl peroxide is required to form hybrid polymers. These OCL-derived hybrid polymers have been applied to various fields such as optics [90,91], anti-corrosion coatings [92], etc. The preparation conditions and essential structural elements that have been used in some applications are summarized in Table 2. For example, Amberg-Schwab et al. studied Si-O-Zr and Si-O-Al ceramic composites modified with methacrylic acid (a network-forming OCL) for transparent barrier coatings. After hydrolysis and polycondensation, the curing process was carried out under UV with 1-hydroxycyclohexylphenylketone as the curing agent [90]. These examples show that the crosslinking of the OCL and other organic modifiers is an essential structural element when utilizing network-forming ligands. Furthermore, the examples presented in Table 2 provide ample evidence of the universality of the structural building blocks of OCL-derived inorganic–organic composites, serving as a template for applications in membrane development, which are scarce or even non-existent.

The introduction of reactive units on both the Si precursor and OCL introduces some complexity around the roles of each structural element in determining the overall characteristics and performance of membranes derived from network-forming organic modifiers. Our previous work attempted to explore this complexity [93]. Figure 13 shows a schematic representation of different possible formation routes utilized in studying the effects of important structural elements. First, the choice of ligand either limits the organic reactive species (an unsaturated C=C bond in this case) to the Si precursor, or both the Si precursor and OCL possess the polymerizable C=C bond. This allows us to study the effect of the ligand bearing a network-forming unit or otherwise. Furthermore, the choice of crosslinking or non-crosslinking allows us to study the effect of the polymer network on the resulting composite. Serving a dual purpose, 3-methacryloxypropyltriethoxysilane (MAPTMS) is the precursor for the siloxane network and possesses a reactive organic modifier (methacrylate group). Two chelating ligands, allyl acetoacetate (AAA) and acetylacetone (AcAc), are instrumental in modifying the zirconium n-butoxide (the transition-metal oxide precursor).

### 4.1. Effect of Organic Chelating Ligand Type

The distinctive difference between AcAc and AAA is the presence of an unsaturated C=C bonding in the AAA ligand, as shown in Figure 13. However, common to both systems is the MAPTMS Si precursor possessing its own unsaturated methacrylate C=C bond. Since the AcAc chelating ligand has no reactive units for crosslinking, homopolymerization of only the MAPTMS moiety can be expected. On the other hand, the AAA possessing polymerizable C=C units can undergo copolymerization with the MAPTMS methacrylate unit.

Although the chemistry of copolymerization of MAPTMS and AAA is not entirely understood, side homopolymerization reactions of only AAA moieties are not very likely. Homopolymerization of AAA has been reported to proceed via a mechanism known as “degradative monomer chain transfer” and found to lead to short-chain polymers of a medium molecular weight. However, Grzybowska et al. found that pairing AAA with styrene generates a copolymer with a high molecular weight [94]. This means that AAA by itself does not propagate a polymerization reaction effectively. As a result of this, some homopolymerization side reactions of AAA might have occurred, but the degradative monomer chain transfer mechanism means that the extent was limited. Matsumoto and coworkers [95] noted that the presence of hydrogen atoms on the carbon adjacent to the double bond (i.e., the allylic double bond) is responsible for the degradative chain transfer mechanism. The resulting monomeric allyl radical is less reactive and has little tendency to initiate a new polymer chain because of a self-stabilizing resonance, which essentially serves as a termination reaction [95]. Consequently, it can be concluded that copolymerization between MAPTMS and AAA was the far dominant reaction since the presence of MAPTMS radicals means that the AAA radicals can be propagated in copolymerization, which is similar to the AAA-styrene copolymer case [93].

Figure 14a shows a schematic image illustrating the network formation in inorganic–organic hybrid polymers depending on the type of organic chelating ligand used in our particular case [93]. In the AAA-derived hybrid polymer, AAA is assumed to have crosslinked with MAPTMS in the copolymer chain, leading to a well-defined microstructure. In the AcAc-derived hybrid polymer, on the other hand, only crosslinking of MAPTMS must have occurred, leaving the AcAc chelates to freely occupy the network structure. The nitrogen adsorption isotherms measured for AAA- and AcAc-derived hybrid polymer powders were compared to ascertain this theory, which is shown in Figure 14b. It is obvious that the AAA-derived hybrid polymer showing a higher amount of adsorbed nitrogen possesses a greater surface area and pore volume compared to the AcAc-derived hybrid polymer. Moreover, at very low relative pressure ranges (P/P_0_: 0–0.001), the steep adsorption of nitrogen for the AAA-derived hybrid polymer means more well-defined micropores compared to the AAA-derived hybrid polymer. Therefore, crosslinking of the organic modifier on the Si precursor was not enough to generate well-defined microstructures compared to copolymerizing with a crosslinkable OCL.

### 4.2. Effect of the Crosslinking of Reactive Organic Groups on Membrane Properties and Separation Characteristics

As highlighted previously, the crosslinked polymer network is a critical structural element of OCL-derived inorganic–organic composites. In the presence of radical initiators, the curing or crosslinking of the reactive organic species can proceed via light or temperature. The vast majority of works utilize UV light in the curing process, as the references in Table 2 show, because the curing process is much faster and leads to very high turnovers, which is important for other application fields. However, due to the importance of maintaining network tunability for molecular separations, the ability to control the curing process with a slower radical initiation and chain propagation process is important. In our study [93], 2,2′-azobisisobutyronitrile (AIBN) was chosen as a thermal radical initiator because it is known to thermally dissociate at temperatures between 60 and 80 °C, which makes it suitable for AAA copolymerization reactions [94,96]. As illustrated in Figure 15a for the AAA-derived hybrid polymer, the state of the structure before crosslinking allows the AAA chelated to Zr and the methacrylate unit of the silane to freely occupy the intra-network spaces. Therefore, structural and performance differences are expected to exist between crosslinked and non-crosslinked polymer membranes.

Figure 15a,b show the cross-sectional morphologies of asymmetric non-crosslinked AAA-derived and crosslinked AAA-derived hybrid polymer membranes, denoted as NC-AAA and CL-AAA, respectively. These images present a clear distinction between the active separation layers comprising the inorganic–organic hybrid material and the substrate layers. As evident from the SEM images, the active top separation layers were successfully fabricated to be less than 50 nm in thickness. This is instructive since controlling the fabrication of membranes that are as thin as possible reduces permeation resistance while simultaneously maintaining the molecular-sieving property at a high level. By contrast, a crack-free, free-standing inorganic–organic hybrid membrane can be very thick at 100 µm and have a low H_2_ permeance of 1.14 × 10^−13^ mol m^−2^ s^−1^ Pa^−1^ [97].

As Figure 15a clearly shows, the CL-AAA membrane formed a distinct layer of about 20 nm at its thickest point over the substrate layer. On the other hand, the NC-AAA membrane formed a thinner, almost indistinct, layer of less than 15 nm at its thickest point on the substrate layer. This can be explained by the fact that the CL-AAA has a much higher polydispersity [93] with an extensively crosslinked structure that is deposited much thicker and distinctly over a porous surface. When the NC-AAA is coated over a porous surface, its much lower polydispersity [93] and non-crosslinked structure allows the particles to settle into the underlying surface, thereby forming a thinner layer that is almost indistinct. Figure 15c,d support this explanation by comparing the deposition behaviors of CL-AAA and NC-AAA on a copper grid, which was observed by transmission electron microscopy (TEM). As shown in these images, the CL-AAA with its extended crosslinked structure was deposited across the grid supports and spaces while the NC-AAA was deposited only on the grid supports and penetrated the spaces. Due to this difference in morphology, it is therefore expected that both non-crosslinked and crosslinked membranes exhibit different microstructural properties.

The microstructural properties of the NC-AAA and CL-AAA membranes were analyzed by estimating their mean pore diameters at different temperatures by using the modified gas translational model. The modified gas translation model proposed by Lee et al. [98] presents a way to estimate a mean pore size of <1 nm for membranes by employing a normalized Knudsen permeance (NKP) method. In formulating the equation for microporous molecular-sieving membranes, the membrane configurational parameters such as porosity, tortuosity, thickness and mean pore diameter should be considered. However, for simplicity, the permeating molecules are assumed to permeate through cylindrical channels, and the proposed modified gas translational model is given in Equation (3) below.
(3)$$P_{i}=\frac{1}{3}\frac{\varepsilon }{\tau L}\left(d_{p}-d_{i}\right)\frac{{\left(d_{p}-d_{i}\right)}^{2}}{d_{p}^{2}}\sqrt{\frac{8}{\pi M_{i}RT}}exp\left(-\frac{E_{p,i}}{RT}\right)$$
where the combination of configurational factors of the membrane and permeating molecule is expressed as porosity ε, tortuosity τ, membrane thickness L, mean pore diameter $d_{p}$ and kinetic diameter $d_{i}$.$E_{p,i}$ is the apparent activation energy of permeation for a gas species i, $M_{i}$ is the molecular weight of the gas species, R is the universal gas constant and T is the permeation temperature. The NKP is the ratio of experimentally obtained permeance of i to the Knudsen permeance of i that can be expected based on a standard gas. In combination with Equation (3) and applying further simplifications, the mean pore diameter $d_{p}$ can be estimated at any permeance temperature T by using Equation (4) expressed as a function of the kinetic diameter $d_{i}$ and by employing $d_{p}$ as the fitting parameter.
(4)$$f_{NKP}=\frac{P_{i}}{P_{s}}\sqrt{\frac{M_{i}}{M_{S}}}\approx \frac{{\left(1-\frac{d_{i}}{d_{p}}\right)}^{3}}{{\left(1-\frac{d_{s}}{d_{p}}\right)}^{3}}$$
where S represents the standard gas.

Figure 16a,b show the calculated NKP values plotted as a function of the kinetic diameter for the NC-AAA and CL-AAA membranes, respectively. In Figure 16a, two different mean pore diameters were estimated for the NC-AAA membrane (0.56 and 0.62 nm) at 50 and 200 °C. In contrast, the estimated mean pore diameters for the CL-AAA membrane at 50 and 200 °C were similar at 0.43 and 0.45 nm, respectively (Figure 16b). This indicates that non-crosslink-derived membranes express different pore sizes depending on the permeation temperature. This may be explained by the fact that the non-crosslinked organic moieties that freely occupy the networks in NC-AAA are flexible, and thus vibrate with an increase in temperature. Therefore, at low temperatures the organic moieties are stiff and occupy much of the intraparticle free volume. A high selectivity such as H_2_/SF_6_ ~1000 can be obtained, but at high temperatures their continuous vibrations create a greater free volume for gas permeation [93]. Hence, it is difficult to estimate a single mean pore diameter in cases where organic moieties are not crosslinked and are free to vibrate.

On the other hand, the expression of similar mean pore sizes at both 50 and 200 °C suggests a rigid, non-flexible structure with little vibrational motion as the temperature increases. Therefore, membrane performance may be adequately predicted at reasonably high temperature ranges due to the lack of structural change in the inorganic–organic hybrid polymer membrane. Thus, the respective morphologies of the NC-AAA and CL-AAA membranes seem to correlate with the microstructural characterization. Based on these observations, the next section attempts to situate the microstructural properties of inorganic–organic hybrid polymer membranes vis à vis state-of-the-art materials that exhibit purely rigid and purely flexible microstructures.

### 4.3. Microstructural Analysis of Organic Chelating Ligand-Derived Hybrid Polymer Membranes in Comparison to State-of-the-Art Materials

The microstructures of membranes can be analyzed by examining the relationships between the permeation behaviors of gas molecules. This usually involves correlating the activation energies of permeation and permeabilities of permeating molecules through the microstructures. Due to the difficulties and inconsistencies associated with comparing absolute quantities of the properties in different microstructures, it becomes convenient to define certain intrinsic microstructural properties such as an “intrinsic rigidity” (or flexibility). For membranes with very flexible microstructures such as rubbery polymer membranes, the activation energy of gas permeance increases with the kinetic diameter. In rigid microstructures such as glassy polymer membranes, the gas kinetic diameter has less influence on the activation energies [33].

Based on this, the intrinsic rigidity of a membrane can be defined as the normalized difference between the activation energies of the permeance of a large gas and that of a smaller gas through a membrane. In the current review, H_2_ and N_2_ have been chosen as the permeating molecules so that the intrinsic rigidity can be expressed as 1 − [*E*_p_ (H_2_)/*E*_p_ (N_2_)]. The relative rigidity from membrane to membrane can thus be determined by the value of this expression. Figure 17 illustrates a spectrum of materials defined by their “intrinsic rigidity” as a function of the ratio of activation energy of N_2_ permeation to that of H_2_ (*E*_p_ (N_2_)/*E*_p_ (H_2_)). A correlation was established such that increasingly “flexible” membranes show increasingly rubbery polymer-like characteristics. On the other hand, increasingly “rigid” membranes show increasingly silica-like microstructures. Organosilica membranes are regarded as rigid, porous membranes with some degree of flexibility depending on the bridging organic group, and the greater the length of the organic C-C bridge between Si atoms, the more flexibility the membrane will demonstrate [99,100]. In Figure 17, it is shown how a BTESO-derived membrane with an octyl hydrocarbon bridge tends to have a more flexible and dense structure compared with a BTESE-derived membrane with an ethyl hydrocarbon bridge.

The NC-AAA membranes show a trend closer to polymer membranes, confirming the theory of flexible non-crosslinked moieties. The CL-AAA membranes, on the other hand, tend to have more “rigid” structures. Therefore, inorganic–organic hybrid materials display different microstructural characteristics depending on the crosslinking state of the organic chelating ligands and other organic modifiers present. These observations present a framework for designing functional molecular-sieving networks such that a temperature-dependent functional group can be incorporated without having to account for the temperature-dependent mean network pore size.

## 5. Carbon–Ceramic Composite Membranes from Organic Chelating Ligand-Derived Composites

Carbon molecular sieve membranes with very good separation abilities are fabricated via the pyrolysis of pure organic polymer materials (hollow fibers, films, coatings, etc.). In a similar fashion, composite carbon molecular sieve membranes have been developed via the pyrolysis of inorganic–organic hybrid polymers such as polydimethylsiloxane (PDMS) blended with other polymers [101], mixed-matrix membranes [102] and Si-C-type membranes derived from the pyrolysis of allylhydridopolycarbosilane (AHPCS) [103] and polytitanocarbosilane (PTCS) [104] preceramic precursors. All these approaches rely on the use of existing and available polymer precursors. However, this section aims to briefly look at carbon–ceramic composite membranes derived from organic chelating ligand-derived inorganic–organic composites.

### 5.1. Carbon–Ceramic Membranes from Network-Modifying Ligand-Derived Composites

The presence of organic modifiers in OCL-derived composites presents an opportunity to transform such hybrids into carbonized networks by pyrolyzing the chelating ligands into a carbon phase. This method of forming carbon–ceramic composites is a novel idea that has been rarely studied. A SiO_2_-ZrO_2_-AcAc composite derived from a network-modifying ligand (AcAc) was pyrolyzed in inert atmospheres to form carbonized SiO_2_-ZrO_2_ (C-SiO_2_-ZrO_2_), as reported in our previous works [105,106]. C-SiO_2_-ZrO_2_ membranes were studied for their H_2_ separation performance and to understand the microstructural transformation necessary for such a performance to be possible. Figure 18a illustrates a theoretical pyrolysis and network formation pathway. AcAc ligands occupying the intra-network spaces undergo pyrolysis under inert conditions at elevated temperatures, which is accompanied by the release of volatile species leaving behind carbon nanoparticles. TEM images of the carbonized SiO_2_-ZrO_2_ particles formed at 550 °C reveal dark patches of ~5 nm or less dispersed in the SiO_2_-ZrO_2_ matrix, as shown in Figure 18b. This provides physical evidence of carbon being present as nanoparticles disperse into the SiO_2_-ZrO_2_ matrix after inert carbonization. Duke and coworkers also studied the carbonization of surfactant-templated silica (hexyl trimethyl ammonium bromide-silica) and found that carbon nanoparticles were dispersed in the silica microstructure [107].

The resulting C-SiO_2_-ZrO_2_ membranes showed different permeation properties depending on the atomic ratio of Si to Zr in the ceramic composite, and can therefore serve different separation applications. As TEM in Figure 18b shows, the C-SiO_2_-ZrO_2_ composite contained carbon nanoparticles which were characterized as having graphitic properties [108], that is, they possess weak van der Waals forces produced by delocalized π-orbital electrons (π–π stacking) and sp^2^-hybridized C-C σ-bonding in the horizontal direction, as illustrated in Figure 19 [105]. The delocalized electrons could specifically interact with polar CO_2_ molecules, allowing for a C-SiO_2_-ZrO_2_ membrane having the molar ratio of Si/Zr = 5/5 to have a unique permeation property, as it is rich in residual carbon.

Figure 20 shows the single-gas permeance for C-SiO_2_-ZrO_2_ membranes (Si/Zr = 5/5 and 9/1), which were both fabricated at 550 °C, as a function of the kinetic diameter of different gases (He (0.26 nm), H_2_ (0.289 nm), CO_2_ (0.33 nm), N_2_ (0.364 nm), CH_4_ (0.38 nm), CF_4_ (0.48 nm) and SF_6_ (0.55 nm)) measured at 300 °C. For a membrane with a Si/Zr ratio of 5/5 that is rich in residual carbon, it was noted that the values for the gas permeance of CO_2_ (1.7 × 10^−10^ mol m^−2^ s^−1^ Pa^−1^) were lower than all the other gases tested. This was attributed to the electronic trapping of CO_2_ between the graphitic carbon layers, which allowed for an interesting high-temperature separation performance in binary H_2_/CO_2_ systems comparable to state-of-the-art molecular-sieving membranes [105].

When the Si content in the C-SiO_2_-ZrO_2_ membrane is increased so that the Si/Zr ratio becomes 9/1, the microstructure of the resulting carbonized ceramic composite becomes rich in Si atoms and lean in residual carbon. As shown in Figure 20, the C-SiO_2_-ZrO_2_ membrane with a Si/Zr ratio of 9/1 has a H_2_ permeance of 16 × 10^−7^ mol m^−2^ s^−1^Pa^−1^, which is 10 times higher than that of the 5/5 membrane (3 × 10^−7^ mol m^−2^ s^−1^Pa^−1^), while also showing a higher H_2_/CH_4_ permeance ratio of 148 compared to 83 [106].

### 5.2. Carbon–Ceramic Membranes from Network-Forming Ligand-Derived Composites

As discussed in the previous section, organic chelating ligands that are network modifying such as acetylacetone can be used for the carbonization of ceramic composites, but due to the low decomposition temperature, carbon–ceramic composites cannot be formed at temperatures much higher than 550 °C. This shortcoming has motivated the research into the possibility of forming carbon–ceramic composites at pyrolysis temperatures up to 700 or 800 °C. The objective is to form composite carbon–ceramic membranes that are similar to the carbon-SiO_2_ membranes derived from conventional polymers, such as polydimethylsiloxane (PDMS) studied by Lee et al. [101] or Si-C membranes from allylhydridopolycarbosilane (AHPCS) [103] and polytitanocarbosilane (PTCS) [104] studied by Wang et al. These conventional composite carbon–ceramic membranes can be fabricated by pyrolyzing their respective inorganic–organic hybrid polymers at temperatures up to 800 °C. This section reviews the formation of a carbon-SiO_2_-ZrO_2_ composite from a network-forming and thermosetting OCL, referred to as a benzoxazine.

Benzoxazines are a family of organic compounds whose basic structure comprise a benzene ring fused to an oxazine ring. Variants of these benzoxazines possess a nucleophilic carboxylic group that can modify transition-metal precursors via chelation, and thus, present an opportunity for the formation of an organic–inorganic composite. The chelated benzoxazine can then polymerize via a ring-opening mechanism into phenolic chains upon thermal exposure or in the presence of cationic radical initiators [109]. In our previous work, we successfully formed a SiO_2_-ZrO_2_-polybenzoxazine (SZ-PB) preceramic resin via several precursors [110,111]. The Si precursor vinyltrimethoxysilane (VTMS), Zr precursor zirconium butoxide (ZrTB) and the carboxylic benzoxazine ligand 3-(3-oxo-1,4-benzoxazin-4-yl) propanoic acid (BZPA) were used as key structural elements in the formation of the thermosetting inorganic–organic hybrid polymer resin. Thermal curing was applied at 90 or 200 °C after hydrolysis and condensation in the presence of a dibenzoyl peroxide (BzO_2_) radical initiator to obtain a SiO_2_-ZrO_2_-polybenzoxazine resin.

Unlike other types of ligands, the polybenzoxazine is expected to form integrated carbonized structures instead of discrete free carbon nanoparticles. The formation of sp^2^-natured graphitic carbon during pyrolysis at high temperatures was confirmed by an X-ray photoelectron spectroscopy (XPS) examination of the cured SZ-PB resin and the derived carbon–ceramic composites after calcination at various pyrolysis temperatures. The narrow C 1s spectra results show spectral peak shifts from 284.4 to 283.2 eV with an increase in pyrolysis temperature up to 850 °C, as presented in Table 3. This was attributed to the transformation of carbon from its sp^3^ form to the graphitic sp^2^ form. According to the respective proportions of the sp^3^ and sp^2^ states, carbon was predominantly present in the sp^3^-hybridized state for the freshly cured resin at 90 °C, as indicated by a proportion of 62% with no detection of sp^2^ carbon. The appearance of sp^2^-hybridized carbon started at 550 °C, and at 750 °C a dramatic change occurs as the sp^3^ carbon was drastically reduced from 53% to 29% and a high proportion sp^2^ carbon of 57% evolved and 58% at 850 °C. Consequently, the sp^3^/sp^2^ carbon ratio was reduced as the calcination temperature increased, although even at 850 °C, the sp^3^/sp^2^ ratio value indicates that the sp^3^ form of carbon persisted, which can be assigned to Si-C bonds from the VTMS.

Figure 21a shows the TEM image of the carbon–ceramic composite generated at 850 °C, which reveals the appearance of the short-range ordered structures of carbon. Although the continuous ordered structures of graphitic carbon could not be observed due to the coexisting amorphous phase of SiO_2_-ZrO_2_, very short patterns representing the ordered structure of the graphite-like layers are apparent. Furthermore, the carbon-SiO_2_-ZrO_2_ (C-SZ) carbon–ceramic membranes prepared on prefabricated substrate layers at 850 °C are shown in Figure 21b via a scanning electron micrograph (SEM) image and energy dispersive spectroscopy (EDS) of the cross-section. The EDS spectrum of C K shows a high concentration of carbon within 500 nm of the top surface of the asymmetric membrane. This shows that the fabrication of continuous ultrathin layers of carbon-SiO_2_-ZrO_2_ was possible.

#### 5.2.1. Effect of Curing Temperatures on Microstructural Properties

The polymerization of benzoxazines is commonly accomplished by organic radical initiation, cationic radical initiation and thermal polymerization [112]. Thermal polymerization of benzoxazine was applied in the formation of a SiO_2_-ZrO_2_-polybenzoxazine hybrid. With the help of a radical initiator, the oxazine ring opens to form a tri-substituted benzene ring. Subsequent thermal applications result in a curing process involving the crosslinking of the opened oxazine and the vinyl groups of the VTMS. This formation model was found to be dependent on the curing temperature [111]. Figure 22 shows the FTIR spectra of a SiO_2_-ZrO_2_-polybenzoxazine resin before and after thermal curing at 90 and 200 °C. The FTIR spectra (red line) revealed that the 90 °C-cured resin showed a spectrum almost similar to that of the fresh as-prepared sample, with the only difference being the disappearance of the BzO_2_ radical initiator peaks. However, after curing at 200 °C the FTIR spectra reveal the disappearance of certain peaks and the appearance of new ones, which can be attributed to a change in the bonding structures as a result of curing. Thus, the observable curing of the vinyl and oxazine moieties occurs at a significantly higher temperature.

Since it has been established that the extent of curing is dependent on the curing temperature, the resulting carbon-SiO_2_-ZrO_2_ composites should also differ in their microstructural properties based on the curing temperature of the precursor SiO_2_-ZrO_2_-polybenzoxazine. The comparison of microstructural properties can be carried out via nitrogen adsorption and evaluation of the membrane permeation behaviors. Figure 23 shows the comparison of N_2_ adsorption–desorption isotherms of carbon-SiO_2_-ZrO_2_ powders derived from 90 and 200 °C-cured SiO_2_-ZrO_2_-polybenzoxazine resins after pyrolysis at 750 °C. Obviously, the composite derived from the SiO_2_-ZrO_2_-polybenzoxazine resin cured at 200 °C showed a microporosity with a type I isotherm compared to that derived from the 90 °C-cured resin. The generated microporosity could be due to micropore templates formed in hyper-crosslinked polymer chains that are transferred to the carbonized composite, which should be beneficial for molecular sieving.

The comparison of the membrane permeance properties supports the findings from the observation of nitrogen adsorption. As shown in Figure 24, carbon-SiO_2_-ZrO_2_ membranes fabricated at 750 °C from SiO_2_-ZrO_2_-polybenzoxazine resins cured at 90 and 200 °C showed different kinetic diameter dependences for single-gas permeance. For the membrane derived from the 90 °C-cured resin, a small cut-off between H_2_ and CH_4_ was obtained at a H_2_/CH_4_ permeance ratio of approximately 11. Nonetheless, the membrane showed respectable H_2_/CF_4_ and H_2_/SF_6_ permeance ratios of approximately 1800 and 11,000, respectively [110]. Due to the superior molecular-sieving ability of the micropores in the membrane derived from the 200 °C-cured resin, a higher H_2_-CH_4_ cut-off was recorded at more than 1000 [111]. Therefore, the curing temperature for thermally crosslinked inorganic–organic hybrid polymers must be carefully considered.

#### 5.2.2. Carbon–Ceramic Composite Membranes’ Performance under Hydrothermal Conditions

The stability of membrane performance under hydrothermal conditions, especially at high temperatures, is one of the major problems plaguing membrane science and technology. Lin et al., for example, reported that SiO_2_ membranes experienced losses of 48 and 77% in regard to the specific surface area and pore volume, respectively, after exposure to 50 mol% of steamed atmosphere at 600 °C [112]. These structural changes are deemed to be the result of the rearrangement of silica networks initiated via steam hydrolysis to form -Si-OH groups and their subsequent recondensation [113,114]. Efforts in recent decades for the adoption of membranes as reactors for steam reforming reactions through process intensification have put the problem into perspective. Duke et al. demonstrated the stable performance of carbonized template silica membranes in the separation of a simulated reformate gas mixture under a hydrothermal condition [107]. They concluded that, while the silica superstructure retained its core hydrophilicity, and thus, its ability to easily form -Si-OH groups in steam, the presence of carbon nanoparticles prevented the formation and migration of silanol groups responsible for the recondensation and densification of traditional silica under hydrothermal conditions [107].

In our previous work, the stability potential of SiO_2_-ZrO_2_-polybenzoxazine-derived carbon-SiO_2_-ZrO_2_ membranes under hydrothermal conditions was demonstrated. Hydrothermal stability experiments were carried out with H_2_O+N_2_ mixtures (steam partial pressure of 90 and 150 kPa) fed to the membranes prepared at 750 °C and 500 °C for several hours [111]. Dry permeance evaluated before and after each hydrothermal condition showed that the permeance values for He and N_2_ (5.2 × 10^−7^ and 6.6 × 10^−9^ mol m^−2^ s^−1^ Pa^−1^, respectively) were unchanged compared with that before hydrothermal stability testing at a steam partial pressure of 90 kPa. After further exposure at a 150 kPa steam partial pressure, the permeance values for He and N_2_ (5.7 × 10^−7^ and 7.5 × 10^−9^ mol m^−2^ s^−1^ Pa^−1^, respectively) were only slightly increased, displaying the robustness of the inorganic–organic hybrid polymer-derived carbon–ceramic composite membranes. It should be noted that the overall pore size distribution of the membrane was retained after the hydrothermal tests. The mechanism of such stability was further investigated by exposing films to similar hydrothermal conditions and evaluating the surface chemical structural difference before and after the steam exposure. Figure 25a shows the narrow C 1s XPS spectra of the carbon-SiO_2_-ZrO_2_ film before and after steam treatment. The prominent peak at 750 °C is sp^2^ carbon, as was previously established [110]. After the exposure to steam, similar proportions of sp^2^ and sp^3^ carbon were retained compared with those before steam treatment. This shows a resistance to the hydrolysis-recondensation mechanism that has been suggested for pure ceramic materials. Upon exposure to a steam atmosphere, free sp^2^ carbon, being stable, shields the -Si-O-Zr- linkages from a H_2_O attack, as illustrated in Figure 25b.

This hydrothermal stability performance can be compared favorably to pure SiO_2_-ZrO_2_ composite membranes such as those studied by Yoshida et al. [48]. A SiO_2_-ZrO_2_ membrane showed very good hydrothermal stability and H_2_/N_2_ selectivity, but the pore size distribution of the membrane changed considerably with hydrothermal exposure. In another SiO_2_-ZrO_2_ membrane studied by Ahn et al. [50], H_2_ permeance was found to have reduced by 56% after the hydrothermal stability test, although there was a corresponding increase in H_2_ selectivity over N_2_. The retention of the high permeance and pore size distribution of carbon-SiO_2_-ZrO_2_ membranes after hydrothermal exposure suggests a better fit for hydrothermal applications.

## 6. Conclusions and Outlook

The underlying idea in this review is the importance and scope of the organic chelating ligand modification of ceramic composites and the microstructural engineering of such modified composites by choosing specific network-forming ligands and the carbonization of the composites by pyrolysis. Network-modifying organic chelating ligands such as acetylacetone can be useful in providing molecular sieving when chelated into ceramic networks. It has been shown from the different works cited that the intra-network occupation of the ligands leads to molecular sieving, which is useful in gas separation and nanofiltration applications. Furthermore, by employing network-forming organic chelating ligands with reactive units, such as allyl acetoacetate, it is possible to develop new kinds of inorganic–organic hybrid polymers. The ligands serve as both organic network modifiers and network formers. The copolymerization of allyl acetoacetate with methacrylate from the Si precursor can lead to the formation of rigid and predictable molecular-sieving networks.

The formation of new kinds of carbon–ceramic composite membranes was also discussed. Notably, it was found that to generate a stable molecular-sieving carbon–ceramic composite membrane, a thermosetting ligand is necessary in addition to choosing suitable polymerization conditions. The membranes that were formed proved to be robust and applicable under hydrothermal conditions.

From the results presented in this review, it is evident that organic chelating ligands present a very wide field of possibilities for membrane development and applications. The selection of a ligand bearing a functional group and a network-forming moiety in addition to the network-forming Si precursor for network engineering and functional interaction with permeating species could be explored. Possible applications could include the separation of CO_2_ from N_2_ and CH_4_, where the trade-off between CO_2_ permeabilities and selectivities is usually a challenge. Network engineering using organic chelating ligands could expand this possibility further. The presence of a CO_2_-philic functional group could be utilized in improving the transport of CO_2_ while the formation of well-defined networks could serve to selectively reject N_2_ or CH_4_ without limiting the transport of CO_2_.

Another important area of application is the separation of propane and propylene as important petrochemical feedstocks. Due to their similar molecular sizes, the separation becomes challenging. Several membranes have been designed to take advantage of their different structural formulas for shape selectivity. Network-forming organic chelating ligands can be used to tailor such membranes for improved shape selectivity. In addition, functional groups can be introduced to selectively support the transport of one over the other.

Furthermore, the hydrothermal stability of the carbon–ceramic membranes derived from the organic chelating ligand-derived composites positions them as good candidates for membrane reactors for hydrogen production. Membranes present a viable alternative to traditional steam reforming reactions, such as the steam reforming of methane whereby the reaction temperatures reach 600–700 °C. Process intensification can be achieved by utilizing membrane reactors for the steam reforming of methane or ethanol, for example. Due to the simultaneous removal of hydrogen produced from the reaction via the membrane, the overall temperature can be lowered, thereby making it a simple and less energy-intensive process. However, due to the fact that efficient steam reforming reactions require a high steam-to-carbon ratio, the membranes need to be stable under hydrothermal conditions. The carbon-SiO_2_-ZrO_2_ membranes reviewed here are very promising candidates for such applications.

## Figures and Tables

**Figure 1 membranes-13-00390-f001:**
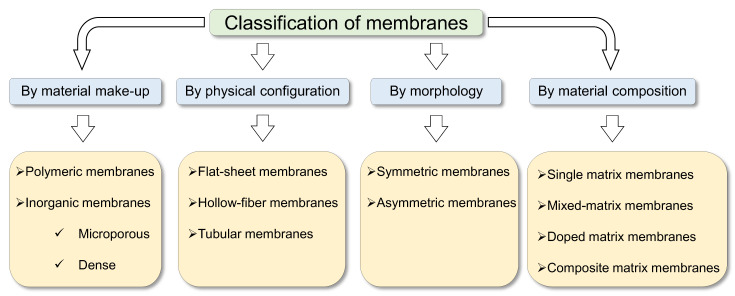
An overview of membrane classification.

**Figure 2 membranes-13-00390-f002:**
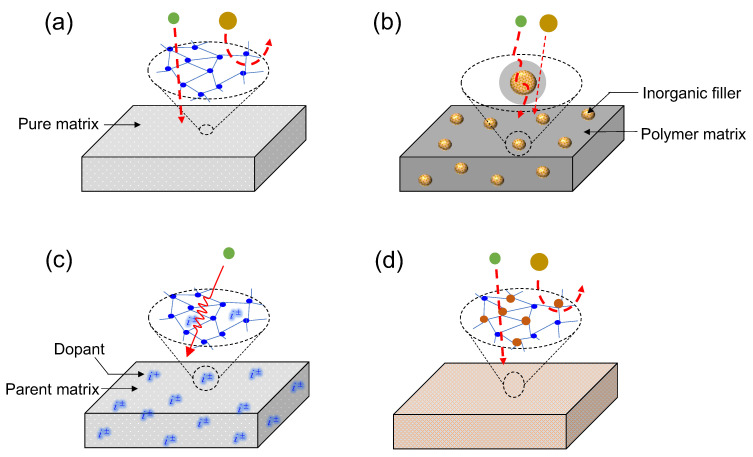
Illustration of the categorization of membranes by matrix composition: (**a**) single-matrix membranes; (**b**) mixed-matrix membranes; (**c**) doped matrix membranes; (**d**) composite matrix membranes.

**Figure 3 membranes-13-00390-f003:**
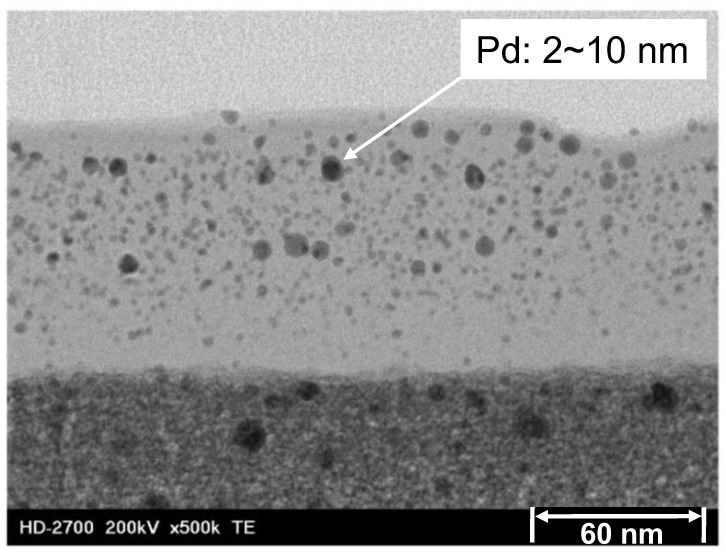
TEM image of the cross-section of a Pd–silica mixed-matrix membrane. Reproduced from [47].

**Figure 4 membranes-13-00390-f004:**
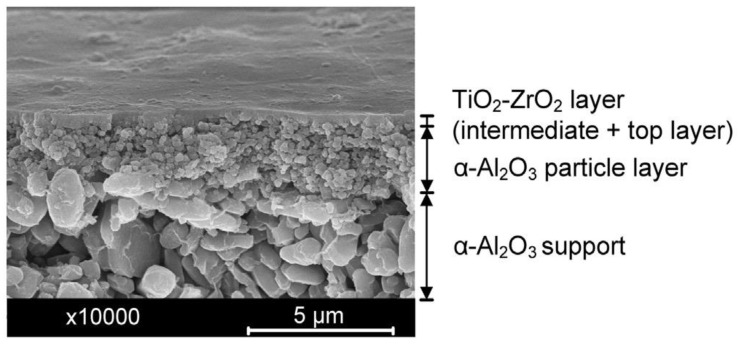
Scanning electron microscopy image of the cross-section of a TiO_2_-ZrO_2_ inorganic–inorganic composite membrane. Reproduced from [51].

**Figure 5 membranes-13-00390-f005:**
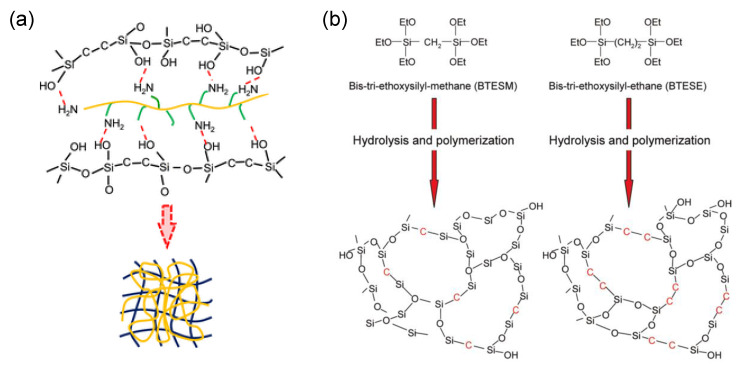
(**a**) Schematic image of organosilica-derived-polymer dual interpenetrating network (IPN). (**b**) Schematic images of amorphous organosilica networks derived from bis(triethoxysilyl)methane and bis(triethoxysilyl)ethane. (**a**,**b**) reproduced from [59,61], respectively.

**Figure 6 membranes-13-00390-f006:**
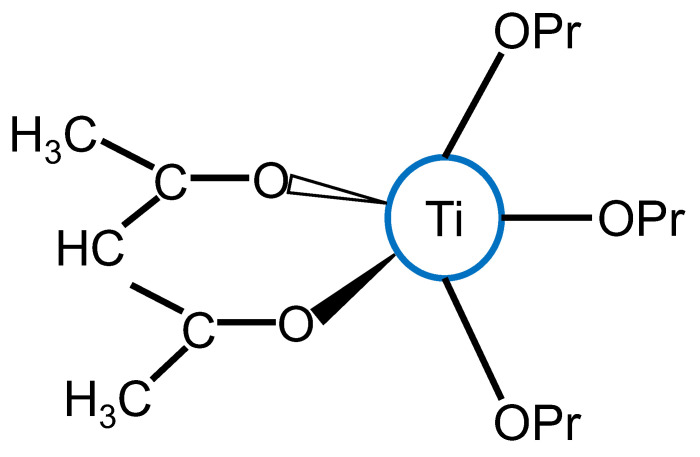
Structural formula of an oligomer formed between a titanium (IV) propoxide and acetylacetone.

**Figure 7 membranes-13-00390-f007:**
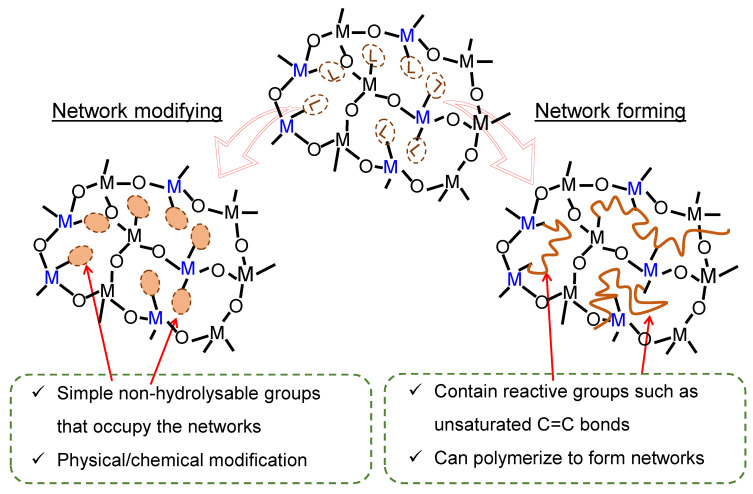
Illustration of network modification and network formation in organic chelating ligand-modified ceramics.

**Figure 8 membranes-13-00390-f008:**
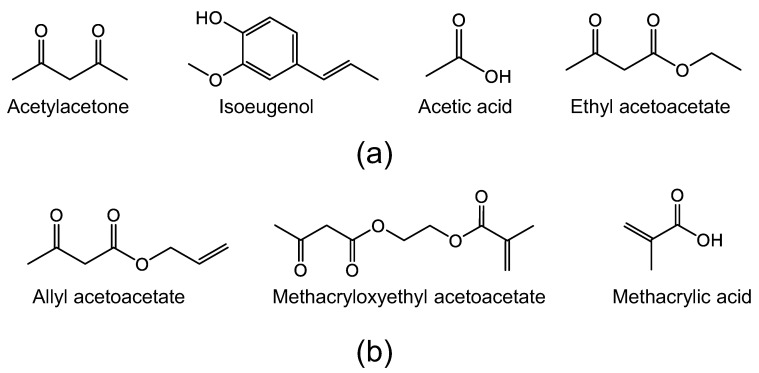
Some organic coordination/chelating ligands: (**a**) network modifiers; (**b**) network formers.

**Figure 9 membranes-13-00390-f009:**
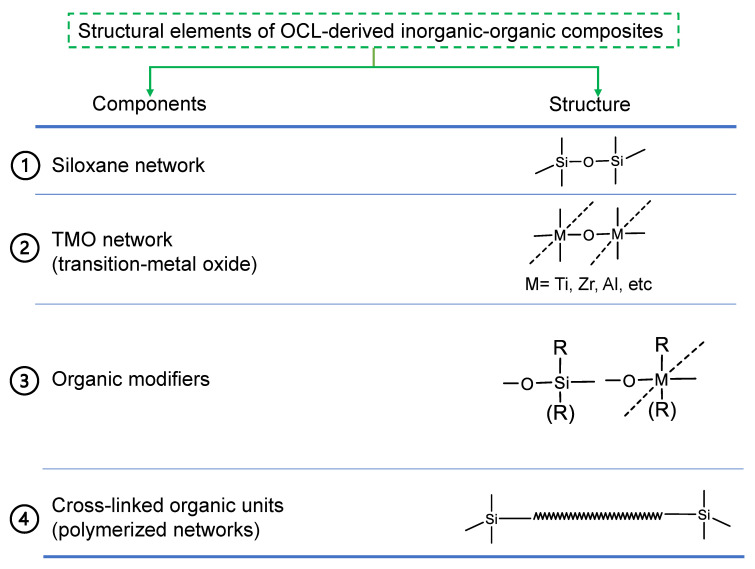
Structural elements in the formation of organic chelating ligand-derived inorganic–organic composites. Adapted from [80].

**Figure 10 membranes-13-00390-f010:**
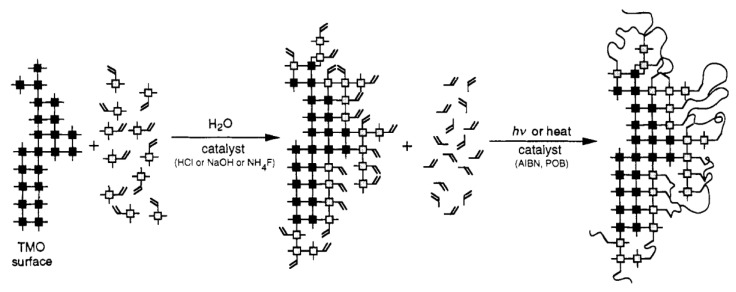
Formation procedures of inorganic–organic composites from organically modified silanes and transition-metal oxides. Reproduced from [82].

**Figure 11 membranes-13-00390-f011:**
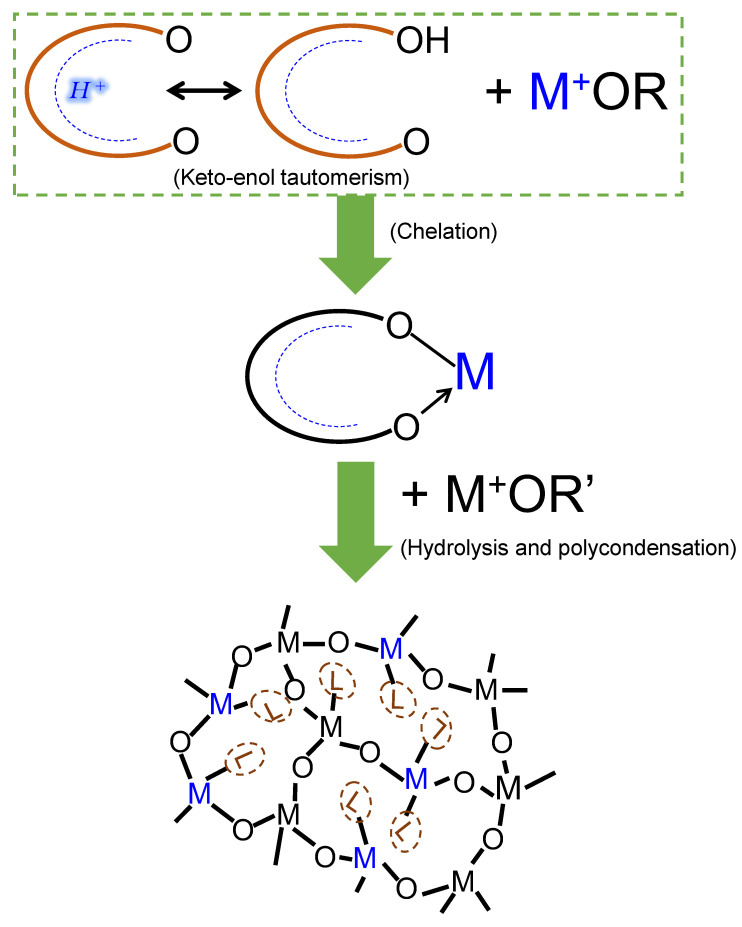
The preparation flow of ligand-modified composite ceramic networks involving a chelation step, followed by the co-hydrolysis and polycondensation step.

**Figure 12 membranes-13-00390-f012:**
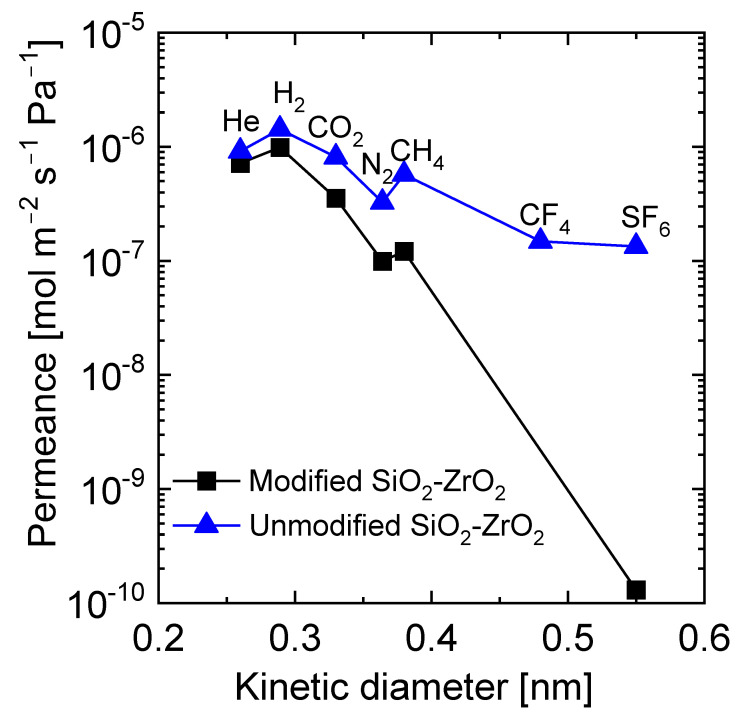
Molecular size dependence of single-gas permeance at 200 °C for SiO_2_-ZrO_2_-AcAc and pure SiO_2_-ZrO_2_ membranes. Adapted from [87].

**Figure 13 membranes-13-00390-f013:**
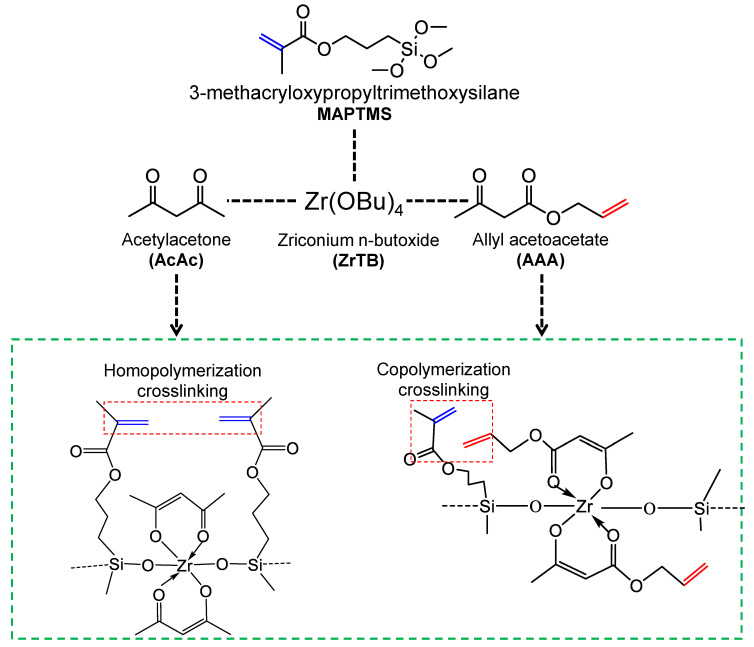
Comparison of the structural elements and formation schemes of different hybrid polymers derived from acetylacetone and allyl acetoacetate ligands.

**Figure 14 membranes-13-00390-f014:**
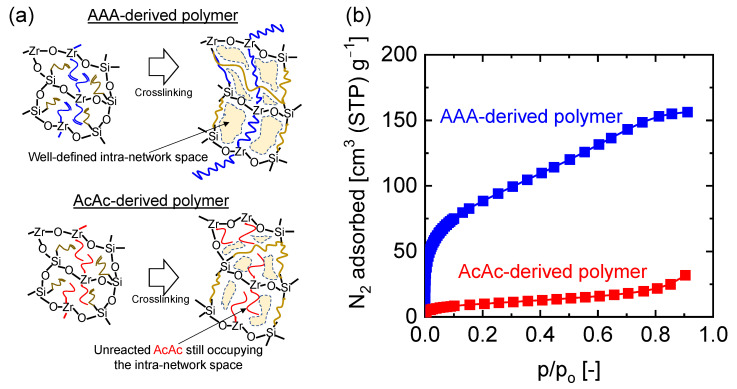
(**a**) Schematic illustration of the network formation in inorganic–organic hybrid polymers based on organic chelating ligand type; (**b**) N_2_ adsorption–desorption isotherms for AAA- and AcAc-derived hybrid polymer powders measured at −196 °C. Results of N_2_ adsorption were adapted from [93].

**Figure 15 membranes-13-00390-f015:**
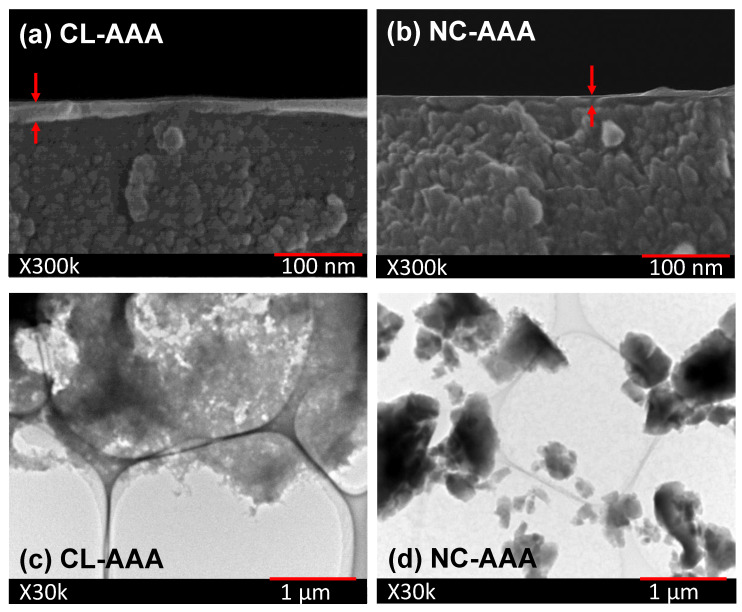
Cross-sectional morphologies of (**a**) CL-AAA- and (**b**) NC-AAA-derived membranes obtained via field-emission scanning electron microscopy (FE-SEM; 300,000× magnification) and transmission electron microscopy (TEM; 30,000× magnification) images of deposited (**c**) CL-AAA- and (**d**) NC-AAA-derived films. SEM and TEM images were adapted from [93].

**Figure 16 membranes-13-00390-f016:**
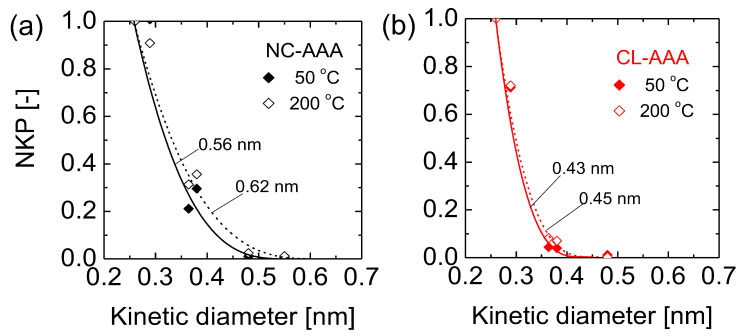
NKP as a function of kinetic diameter for estimating the mean pore size for (**a**) NC-AAA and (**b**) CL-AAA membranes. The solid and dashed lines are theoretical curves. Adapted from [93].

**Figure 17 membranes-13-00390-f017:**
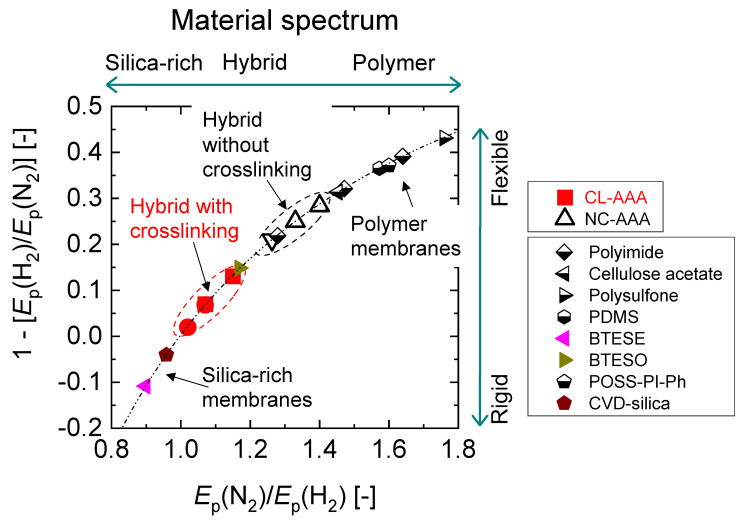
The correlation between “intrinsic rigidity” of 1 − [*E*_p_ (H_2_)/*E*_p_ (N_2_)] and material class calculated based on *E*_p_ (N_2_)/*E*_p_ (H_2_) values for different membranes. Adapted from [93].

**Figure 18 membranes-13-00390-f018:**
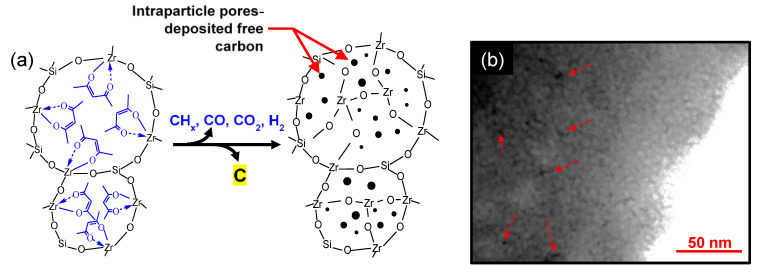
(**a**) Illustration of expected pyrolysis and network formation pathway of SiO_2_-ZrO_2_-AcAc to C-SiO_2_-ZrO_2_; (**b**) TEM image of 550 °C-derived C-SiO_2_-ZrO_2_. Adapted from [105].

**Figure 19 membranes-13-00390-f019:**
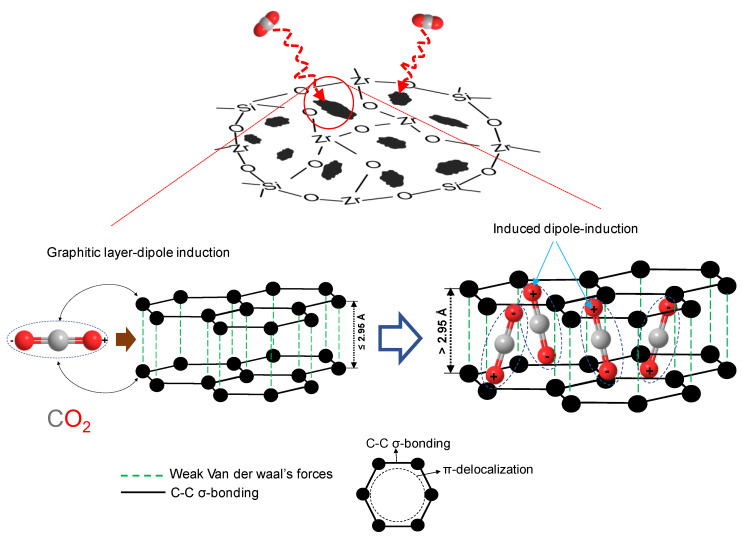
Schematic illustration of the mechanism of interaction between CO_2_ and carbon nanoparticles in C-SiO_2_-ZrO_2_. Reproduced from [105].

**Figure 20 membranes-13-00390-f020:**
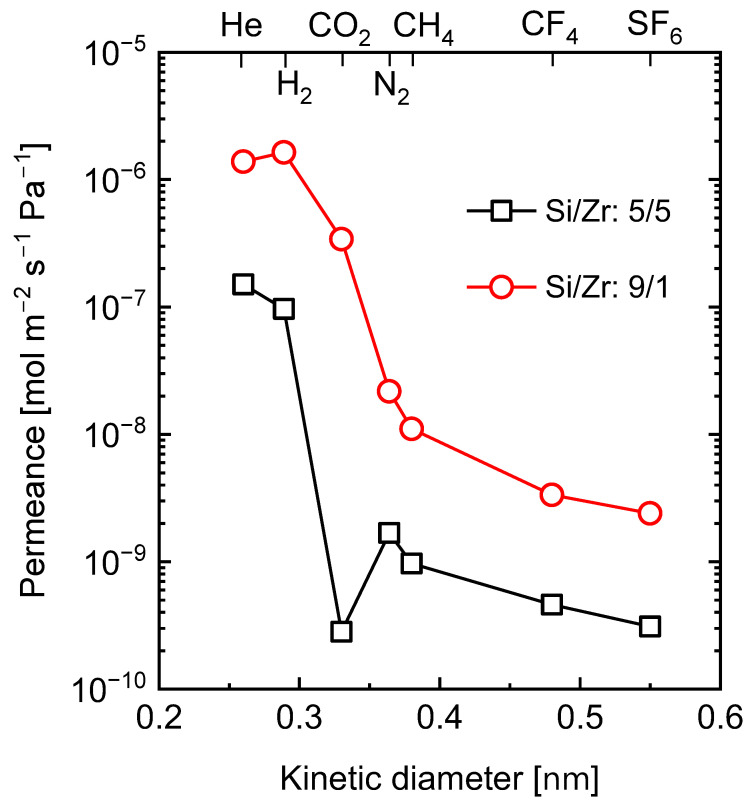
Single-gas permeance at 300 °C as a function of kinetic diameter for C-SiO_2_-ZrO_2_ membranes (Si/Zr = 5/5 and 9/1). Adapted from [105,106].

**Figure 21 membranes-13-00390-f021:**
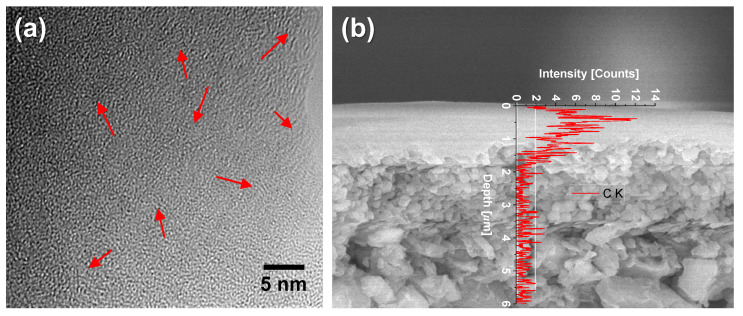
(**a**) TEM image of the carbon–ceramic sample pyrolyzed at 850 °C. Arrows point to short patterns representing the ordered structures of the graphite-like carbon. (**b**) SEM image of a SiO_2_-ZrO_2_-polybenzoxazine-derived carbon-SiO_2_-ZrO_2_ membrane fabricated at 850 °C. Adapted from [110].

**Figure 22 membranes-13-00390-f022:**
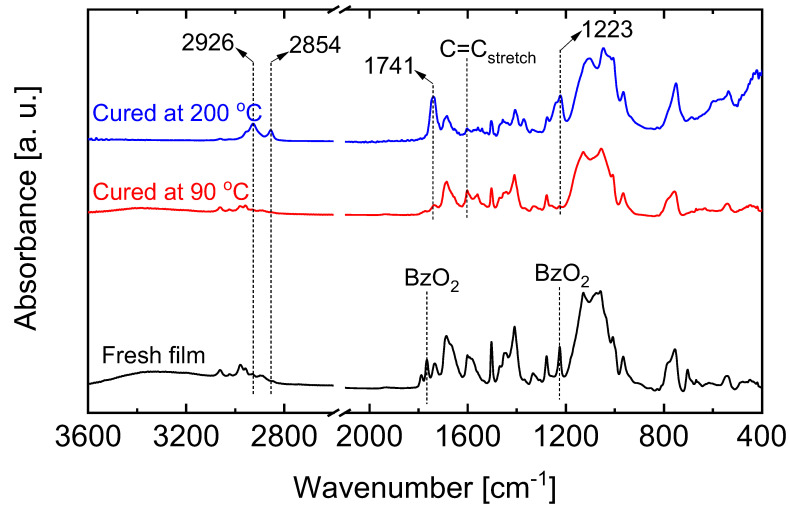
FTIR spectra of SiO_2_-ZrO_2_-polybenzoxazine before and after thermal curing at 90 (red line) and 200 °C (blue line). Reproduced from [111].

**Figure 23 membranes-13-00390-f023:**
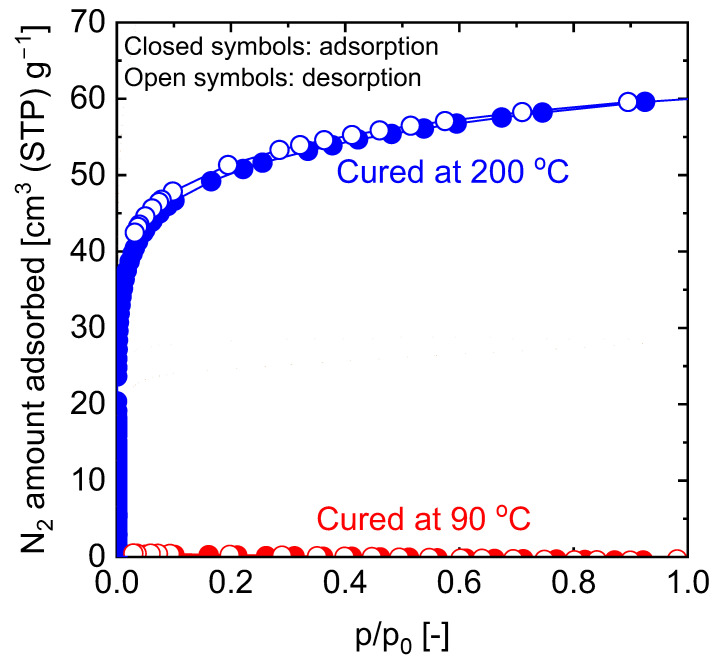
Nitrogen adsorption and desorption isotherms at 77 K of carbon-SiO_2_-ZrO_2_ powders derived from 90 °C- and 200 °C-cured SiO_2_-ZrO_2_-polybenzoxazine resins after pyrolysis at 750 °C. Adapted from [110,111].

**Figure 24 membranes-13-00390-f024:**
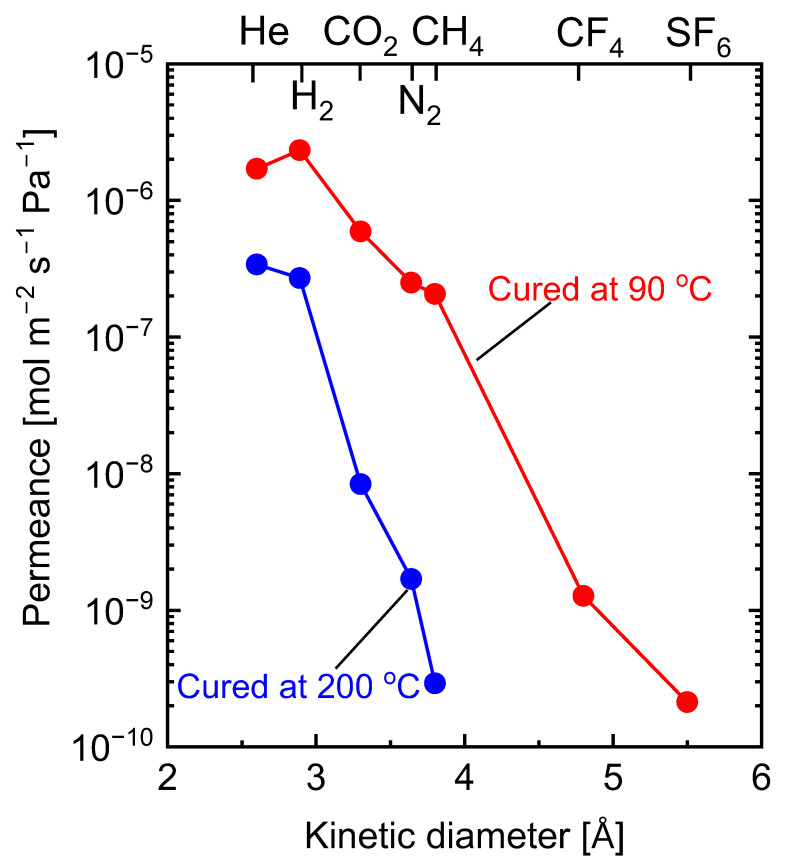
Single-gas permeance at 300 °C as a function of kinetic diameter for carbon-SiO_2_-ZrO_2_ membranes prepared from SiO_2_-ZrO_2_-polybenzoxazine resins cured at 90 and 200 °C. Adapted from [110,111].

**Figure 25 membranes-13-00390-f025:**
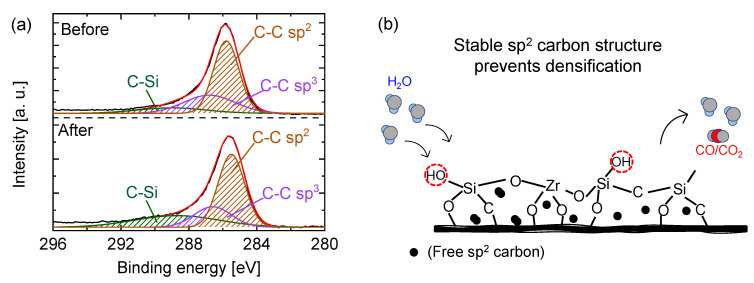
(**a**) C 1s X-ray photoelectron spectroscopy of carbon-SiO_2_-ZrO_2_ films prepared at 750 °C from 200 °C-cured SiO_2_-ZrO_2_-polybenzoxazine resin before and after steam treatment at 500 °C, 90 kPa steam partial pressure. (**b**) Schematic illustration of the microstructural stability of carbon-SiO_2_-ZrO_2_ membranes fabricated upon exposure to steam. Adapted from [111].

**Table 1 membranes-13-00390-t001:** Different ligand-modified composite ceramic membranes and their applications.

Membrane ^†^	Modifying Ligand	Appl.	Permeance [10^−7^ mol m^−2^ s^−1^ Pa^−1^]	Gas pair Selectivity [-]	Water Perm. [LMH bar^−1^]	MWCO [g/mol]	Ref.
TiO_2_-ZrO_2_	Isoeugenol	GS	2.0 (He) 1.4 (CO_2_)	65 (He/N_2_) 46 (CO_2_/N_2_)	-	-	[83]
TiO_2_-ZrO_2_	Diethanolamine (DEA)	GS	1.8 (He) 0.16 (CO_2_)	12 (He/N_2_) 1.1 (CO_2_/N_2_)	-	-	[83]
TiO_2_-ZrO_2_	Diethanolamine (DEA)	GS	3.0 (H_2_)	54 (H_2_/butane)	-	-	[84]
TiO_2_-ZrO_2_	Methyl gallate	GS	8.6 (He)	62 (He/N_2_) 111 (He/CH_4_)	-	-	[85]
TiO_2_-ZrO_2_	Ethyl ferrulate	GS	26.9 (He)	10.6 (He/N_2_) 13.6 (He/CH_4_)	-	-	[85]
SiO_2_-TiO_2_	Acetylacetone	GS	- -	2.43 (He/N_2_) 2.16 (N_2_/CO_2_)	-	-	[86]
SiO_2_-TiO_2_	Isoeugenol	GS	- -	2.39 (He/N_2_) 2.22 (N_2_/CO_2_)	-	-	[86]
SiO_2_-ZrO_2_	Acetylacetone	GS	10 (H_2_)	7600 (H_2_/SF_6_)	-	-	[87]
TiO_2_-ZrO_2_	Ethyl acetoacetate	NF	-	-	5.6	760	[88]
TiO_2_-ZrO_2_	2,3-dihydroxynaphthalene	NF	-	-	9.2	670	[88]
TiO_2_-ZrO_2_	3,5-di-tert-butylcatechol	NF	-	-	-	500	[89]

^†^ All referenced membranes were prepared as asymmetric-supported membranes. Appl.— applications, GS—gas separation, NF—nanofiltration, Water perm.—water permeability, LHM bar^−1^—l m^−2^ h^−1^ bar^−1^, MWCO—molecular weight cut-off.

**Table 2 membranes-13-00390-t002:** Examples of structural elements and preparation conditions and applications of inorganic–organic hybrid polymers derived from network-forming organic chelating ligands.

Author(s)	Inorganic unit	Reactive Organic Species Precursor/Organic Modifier	Hydrolysis/Condensation Conditions	Crosslinking/Curing Conditions	Application(s)
Amberg-Schwab et al. [90]	Si-O-Zr Si-O-Al	✓ Silanes: Methacryloxypropyltrimethoxysilane N-trimethoxysilylpropyl-N,N,N-trimethylammonium chloride ✓ Chelating ligands: Methacrylic acid Triethanolamine	✓ H_2_O ✓ No catalyst ✓ Temperature: 15 °C	✓ Curing agent: 1-hydroxycyclohexylphenylketone ✓ UV curing	Transparent barrier coatings
Le Guevel et al. [91]	Si-O-Ti	✓ Silanes: Methacryloxypropyltrimethoxysilane ✓ Chelating ligands: Acetylacetone Methacrylic acid Trifluoroacetic acid	✓ H_2_O molar ratio: 0.75 ✓ HCl catalyst: 0.01 molar ✓ Temperature: 25 °C	✓ Curing agent: Irgacure184 ✓ UV curing	Optical waveguides
Rodic et al. [92]	Si-O-Zr	✓ Silanes: Tetraethoxysilane Methacryloxypropyltrimethoxysilane ✓ Chelating ligand: Methacrylic acid	✓ H_2_O molar ratio: 2.075 ✓ HCl catalyst: 0.001 molar ✓ Temperature: 25 °C	✓ Curing agent: none ✓ Thermal aging at 100 °C	Anti-corrosion coating

**Table 3 membranes-13-00390-t003:** C 1s XPS elemental analysis of SiO_2_-ZrO_2_-polybenzoxazine preceramic resin after pyrolysis at different temperatures. Adapted from [110].

Pyrolysis Temperature [°C]	C 1s Binding Energy [eV]	Sp^3^ [%]	Sp^2^ [%]	Sp^3^/Sp^2^ [-]
90	284.4	62	-	-
550	284.3	53	6.1	8.7
750	283.3	29	57	0.51
850	283.2	27	58	0.47

## Data Availability

Not applicable.

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
