# Peer review of "A Brief Overview of the Microstructural Engineering of Inorganic–Organic Composite Membranes Derived from Organic Chelating Ligands"

_membranes, 2023, doi:10.3390/membranes13040390_

Round 1

Reviewer 1 Report

This manuscript presented a brief review of the microstructural engineering of inorganic-organic composite membranes derived from the organic chelating ligand. The manuscript is written well and requires some revision.

 1.     Fig.1 shows the classification of membranes, generally, by physical configuration, the membranes are classified into hollow fiber membrane, flat membrane, and tubular membrane; and by morphology and structure, membranes can be classified into asymmetric and symmetric membranes.

2.     In table 1, it was better to present the gas permeance of the membrane and the structure of membranes, asymmetric and symmetric. The development of the composite membrane and asymmetric membrane is required.

3.     In table 2, the inorganic-organic hybrid polymers derived from network forming organic chelating ligands were not used as membranes, but as coating film.

Reviewer 2 Report

This is a sound and interesting review. I recommend publication after addressing the following points;

- Authors should give more details on the chemistry and the mechanism of formation of the discussed material.

-It would be important to discuss the pore size, pore geometry and pore density (pore number) when discussing the porous material.

-The application of the discussed material should be developed and discussed a bit more in details.

Round 2

Reviewer 2 Report

Authors have done a thorough revision and have answered all the comments adequately.